# Emerging Approach for Fish Freshness Evaluation: Principle, Application and Challenges

**DOI:** 10.3390/foods11131897

**Published:** 2022-06-26

**Authors:** Zhepeng Zhang, Ying Sun, Shangyuan Sang, Lingling Jia, Changrong Ou

**Affiliations:** Department of Food Science and Engineering, Ningbo University, Ningbo 315211, China; zzp330683@163.com (Z.Z.); sysy42365@163.com (Y.S.); sangshangyuan@nbu.edu.cn (S.S.); jialingling@jiangnan.edu.cn (L.J.)

**Keywords:** visual indicators, active sensors, spectroscopic techniques, nondestructive, fish freshness

## Abstract

Affected by micro-organisms and endogenous enzymes, fish are highly perishable during storage, processing and transportation. Efficient evaluation of fish freshness to ensure consumer safety and reduce raw material losses has received an increasing amount of attention. Several of the conventional freshness assessment techniques have plenty of shortcomings, such as being destructive, time-consuming and laborious. Recently, various sensors and spectroscopic techniques have shown great potential due to rapid analysis, low sample preparation and cost-effectiveness, and some methods are especially non-destructive and suitable for online or large-scale operations. Non-destructive techniques typically respond to characteristic substances produced by fish during spoilage without destroying the sample. In this review, we summarize, in detail, the principles and applications of emerging approaches for assessing fish freshness including visual indicators derived from intelligent packaging, active sensors, nuclear magnetic resonance (NMR) and optical spectroscopic techniques. Recent developments in emerging technologies have demonstrated their advantages in detecting fish freshness, but some challenges remain in popularization, optimizing sensor selectivity and sensitivity, and the development of algorithms and chemometrics in spectroscopic techniques.

## 1. Introduction

The FAO reports that the demand for fish has increased all over the world in the past few decades. In 2018, global fish production increased to 178.5 million metric tonnes [1]. It is well established that fish provide not only protein and fatty acids, but also vitamin A, vitamin D and a number of trace elements [2]. Owing to the internal and inherent features, such as the brittle muscle tissue, the activity of endogenous protease, as well as improper treatment methods and storage conditions, fish are inclined to corrupt and produce substances threatening human health [3].

Freshness is one of the crucial quality indicators during fish processing, marketing, preservation and consumption. It reflects the degree of physical, chemical, biochemical and microbial changes in fish [4]. Traditional fish freshness detection techniques such as sensory, physicochemical and microbiological require advanced instruments and skilled operators, as well as tedious sample preparation processes. While the importance of the above methods is undeniable, these methods cannot guarantee that the samples will not be spoiled, nor are they suitable for assessing the freshness of fish in real-time. Potential approaches for detecting and evaluating the freshness of fish are, therefore, of fundamental importance.

At present, many methods for assessing the freshness of fish products are emerging. According to the statistics of the Web of Science database, among the 1972 published articles on the subject of “fish freshness”, the research on indicators and spectroscopic techniques accounted for 24.75% and 9.69%, respectively. It is worth noting that the proportion of articles on image processing is 4.87%, indicating that there are few research ideas of this type. On the one hand, it is derived from the indicator in intelligent packaging as an effective pathway for monitoring fish freshness, which displays distinctive advantages, including real-time analysis and nondestructive and cost-effective nature [5]. Indicators contain one or more natural pigments that change color due to volatile odors, which can then be used to report fish freshness. Additionally, packaging materials may respond to triggers by releasing active ingredients such as antioxidants or antimicrobials, which then diffuse and extend the shelf life of the fish. Active sensors are known for their rapid analysis and strong pertinence, and the targets detected by gas sensors and biosensors are mainly amine volatile gases and hypoxanthine generated during fish spoilage, respectively. Electronic sensory techniques mainly rely on non-specific physical adsorption between the analyte and the target molecule, which makes it susceptible to interference from other substances in the environment, especially humidity. In order to facilitate consumers to obtain better feedback on the terminal results, researchers have proposed unique methods and viewpoints, including the use of near-field communication (NFC) in smartphones [6,7]. On the other hand, with the development of spectroscopy and other emerging technologies, a number of nondestructive instruments have been popularized for evaluating fish freshness. These technologies have the advantages of speed, relatively low cost and environmental protection, and provide a large amount of information in one test, making them suitable for on-line process control. Furthermore, spectroscopic techniques typically require little sample preparation and prevent sample damage [8]. In most of the presented papers, chemometric approaches can be applied to data acquired by the electronic and spectroscopic techniques, allowing experimental data to be processed to create qualitative and quantitative models and enabling a rapid and non-invasive measurement process. Unlike previous articles, we comprehensively discussed the integration of visual indicators in intelligent packaging, active sensors, nuclear magnetic resonance (NMR) and optical spectroscopic technologies, and their limitations and future prospects.

In this review, the principle and application of several methods (Figure 1) are introduced in fish freshness evaluation, and we present a series of practical challenges. Firstly, the parameters are described briefly chosen for monitoring the freshness, including sensory, physical, chemical and microbiological attributes. Due to the limitations of traditional methods and the rapid development of sensors, two advanced technologies, visual freshness indicators derived from intelligent packaging and active sensors, are summarized. In particular, the text also covers the application of NMR and optical spectroscopic techniques such as fluorescence, infrared and near-infrared in the detection of fish freshness. Simultaneously, the methods of each technique are fairly unique and different from each other as well as their own advantages and disadvantages. Despite tremendous recent advances in determining the freshness of fish, there is still a need for a more economical and simpler technical means to use today, and research on this is already underway.

## 2. Conventional Freshness Quality Attributes of Fish

### 2.1. Sensory Attributes

Sensory attributes are the principal parameters of fish quality, which are the main parameters that consumers can pay attention to when purchasing fresh fish. Sensory attributes are related to the appearance, color, meat elasticity or texture, odor and taste of fish. Based on these variables, various sensory attributes could be used to classify fish freshness [9]. In addition, the use of any chemical or analytical tools must comply with the results of sensory evaluation [10].

The main method for the comprehensive evaluation of sensory attributes is the quality index method (QIM), based on the appraisement of sensory quality parameters concerning largely the skin, eyes, gills, and abdomen of fish. Developed for each species of fish, this method can assess parameters that change significantly over its shelf life and is widely used to estimate fish quality, although support by other methods is required [11]. QIM provides information about fish freshness status through freshness measurement, as a prediction of the remaining shelf life of fish products. It is based on the interaction between the quantity and quality of the desired fish sample and the deterioration of the fish, so that compliance with the quality requirements can be demonstrated [12]. Nevertheless, the results of the sensory evaluation are quantified inconveniently, and it also has strong subjectivity and one-sidedness. Even if the inspectors have rich experience, it is difficult to give accurate conclusions. Moreover, the complexity of this method is high, which cannot meet the needs of the industry for fast and easy implementation methods.

### 2.2. Chemical Attributes

Fish freshness mainly depends on trimethylamine (TMA) and total volatile basic nitrogen (TVB-N) values, which affect enzymatic action and bacterial degradation. Among them, TVB-N content is widely regarded as the key index for detecting fish spoilage and it is composed of volatile nitrogen compounds such as TMA, dimethylamine (DMA) and NH_3_, which are produced through the hydrolysis of specific amino acids by microorganisms [13,14]. Moreover, the content of TVB-N is directly proportional to the growth of bacteria in the process of fish spoilage [7]. In China, the content of TVB-N below 25 mg/100 g is acceptable. Generally speaking, the determination of TVB-N is determined by steam distillation and the micro diffusion method, but these methods require long-duration preparation and complex operation.

Another chemical indicator to measure fish freshness is the K-value, which represents the whole deterioration cycle of ATP degradation into IMP and hypoxia. It is related to the sensory quality of fish, and a higher K-value indicates a higher rate of ATP decomposition [14]. As for the K-value, it was defined according to the formula [15]:$$\mathrm{K-value}\;(\%)\;=\;\frac{\mathrm{HxR}\;+\;\mathrm{Hx}}{\mathrm{ATP}\;+\;\mathrm{ADP}\;+\;\mathrm{AMP}\;+\;\mathrm{IMP}\;+\;\mathrm{HxR}\;+\;\mathrm{Hx}}\times 100$$

ATP is adenosine triphosphate, ADP is adenosine diphosphate, AMP is adenosine monophosphate, IMP is inosine monophosphate, HxR is inosine and Hx is hypoxanthine.

ATP and its degradation products can be determined by high-performance liquid chromatography (HPLC). Accurate results can be obtained according to the retention time of chromatographic columns. However, in HPLC analysis, sample preparation is cumbersome, the analysis time of each sample is long and the instrument is expensive.

### 2.3. Physical Attributes

Physical attributes, including color and texture such as the shape, size, volume, weight and so on, play a major role in determining fish freshness. The changes in fish surface texture and color directly reflect fish freshness. Originally, the most used tool to detect the color of fish is the chromometer, although now there are many new ways to measure color. Color difference analysis can rapidly detect the freshness of fish; nevertheless, the storage temperature has a great impact on the surface color of fish, which may cause errors. Therefore, the physical evaluation method is only used as an auxiliary method for freshness evaluation.

### 2.4. Microbiological Attributes

In the process of fish spoilage, various microorganisms produce special odor compounds, such as nitrogen compounds, aldehydes, ketones and esters [9]. Therefore, microbial activity is one of the key factors leading to fish spoilage. These approaches could be the quantification during storage of microbial action through total viable counts (TVC) or specific spoilage organisms (SSO). The method of determination is mainly plate colony counting. However, in the initial stage of corruption, microbial indicators cannot accurately discriminate the freshness of fish, and traditional microbial detection requires complicated sample preparation procedures and trained personnel, which cannot monitor the freshness of fish in real-time, nor can it be used for rapid detection.

## 3. Various Sensors for Evaluating Fish Freshness

The sensor is a kind of detection device with extensive coverage and strong applicability. It converts the measured information into electrical signals or other required forms of information output according to certain rules, so as to meet the requirements of information transmission, processing, storage, display, recording and control. Nowadays, various sensors could monitor the freshness of fish in intelligent packaging, and it has good accuracy and sensitivity. However, irreversibility is also a difficult problem that needs to be solved in the application of sensors. The following introduces the visual indicators derived from intelligent packaging and active sensors, respectively.

## 4. Visual Freshness Indicators Derived from Intelligent Packaging

### 4.1. Overview of Intelligent Packaging

Intelligent packaging is a new method developed to meet the growing concerns of consumers regarding food quality and safety. Generally speaking, the design principle of an intelligent packaging system is the interaction between food quality information and the packaging environment to detect changes in the headspace and follows the tracks of product history [16]. To monitor the quality of fish, various types of indicators are applied to intelligent packaging such as time–temperature integrators and freshness sensors, providing real-time monitoring of fish quality to the consumers [17]. Not only can some intelligently packaging monitor food quality, as has been previously reported, but it also delays the growth of microorganisms [18]. The application of intelligent packaging based on pigment and the biopolymer matrix has emerged as an attractive alternative for use in food packaging [19].

### 4.2. Principle of Freshness Indicator

Commonly, visual freshness indicators are composed of two substantial parts, a solid matrix and one or more pigments that are sensitive to pH change [5]. The design of a pH-sensitive intelligent indicator is shown in Figure 2.

During fish storage, volatile amines are released due to microbial activities, and the dye color changes with the pH value of the packaging headspace. Indicator systems provide qualitative or semi-quantitative information through color changes. These intelligent facilities can be attached to the inside of a package so that the freshness of fish could be evaluated in real-time [20]. Therefore, the perceptible color change of the indicators on the packaging has the characteristics of being fast and nondestructive and has wide market prospects.

### 4.3. Pigment

Color is an important sensory attribute that consumers can use to directly evaluate product quality without coming into contact with food [21]. In recent years, synthetic dyes and natural colorants, which can change color with changes in food quality, have been widely used as sensor materials for polymer matrices in intelligent packaging. A summary of natural colorant indicators and synthetic dye indicators for fish freshness monitoring is offered in Table 1.

#### 4.3.1. Synthetic Dyes

Synthetic dyes such as bromocresol purple, bromocresol green (BCG), bromophenol blue and cresol red, coupled with the species-specific variations in indicators, have been used as pH-sensitive pigments. For instance, the increase in microbiota, pH value and amine concentration, combined with the color variation of immobilized BCG from yellow to blue, allow us to easily confirm the deterioration of fish via naked eye observation [37,38,39]. Mo et al. also fixed BCG on porous anode aluminum (PAA) film to fabricate a colorimetric indicator, which can respond to the release of TVB-N during fish spoilage by visible color changes and has the advantages of large surface area and high porosity regularity [33]. Another remarkable finding of this research was the preparation of an indicator via the incorporation of alizarin in starch-cellulose paper [36]. Research showed that the alizarin-starch-cellulose (ASC) indicator had great color efficiency and the color change of ASC from orange to reddish-brown corresponded with the TVB-N content of fish. Notably, there was good compatibility between alizarin and the matrix. The water solubility of the indicator label decreased, but the swelling rate did not change after adding alizarin. In addition, a recyclable indicator based on PANI shows obvious color variation from green to blue and can be recycled at least three times [40]. Ashraf et al. [41] synthesized a new polyaniline-curcumin (CR)-copper-cobalt hybrid composite and evaluated its ability to detect TVB-N content. The results show that the material exhibited excellent response efficiencies to ammonia and mono-, di-, and trimethylamines when evaluated by visual, spectroscopic, and electrochemical methods.

#### 4.3.2. Natural Colorants

The food industry prefers synthetic dyes owing to their high strength, steadiness, multifunction, and low cost. However, consumers prefer natural colorants (plant, fruit and vegetable extracts), because they are worried about the safety of synthetic food colorants [42,43]. It is worth noting that when using colorants for different purposes, these compounds have special biological and environmental characteristics. Owing to their low or non-toxic and eco-friendly nature, convenient preparation, easy access, and renewable and pollution-free characteristics in the food industry, natural colorants have been suggested as a suitable alternative to synthetic dyes [42].

In the natural colorants of fruit resources, anthocyanin (ATH) has notable color changes at different pH values and has great potential in intelligent packaging [44]. Importantly, as a water-soluble pigment, ATH also has interesting biological properties such as antioxidant and antibacterial activity. Studies have shown that packaging films based on blueberry and blackberry pomace integrated with a biopolymer exhibited color changes with pH and also showed high antioxidant potential [6,45]. However, with the increase in pH value, the structure and content of anthocyanin molecules change, which also affects the antioxidant and chromaticity characteristics [46]. Another interesting study is the development of an anthocyanin-based indicator and a mobile phone application by Fang et al. Visual indicators are integrated into a mobile app with the ability to capture images and upload and display real-time results to automate the entire identification process, with a prediction accuracy of 92.6%. The analysis speed of this method is fast, which provides a basis for the development of other chromaticity indicators on mobile phones [47].

Compared with ATH, the color change of CR from yellow to orange is not palpable, thus it is hard to precisely determine fish freshness with the naked eye [48]. In order to address this problem, Chen et al. fixed CR and ATH (ratio 2:8) in a film of polyvinyl alcohol (PVA) and glycerol [22]. This intelligent film can provide three different colors, distinguishing three freshness levels (high-quality freshness, medium freshness and corruption) into semi-quantitative form, which avoided the visual error caused by using only CR.

In addition, Amaranthus leaf extract (ALE) is rich in phenolic compounds and betaland, which has good antioxidant activity. The addition of ALE endows the film with antioxidant, antibacterial and intelligent properties. Kanatt developed a compound intelligent film with PVA and gelatin incorporated with ALE to monitor fish freshness [17]. With the increase in volatile amine, the color of the film changes from red to yellow.

### 4.4. Solid Matrix

In order to develop the functions of these pH-sensitive colorants for intelligent packaging, it is necessary to immobilize these pigments by integrating them into the base material. Typically, the films can be entrenched in packaging materials or attached to the inside of the package, and intelligent packaging systems are able to monitor and provide information about fish freshness. More than that, these composite systems can not only afford a good barrier to block light and moisture, but also have certain mechanical resistance and preservative properties. This can better protect the intelligent film and avoid the impact of high moisture activity. The following subsections introduce three main natural substrates.

#### 4.4.1. Chitosan

Chitosan (CS) is obtained from chitin deacetylation in the cuticle of insects and crustaceans and is the only basic polysaccharide in natural polysaccharides. Various studies show that CS has biocompatibility, certain edibility, fine film formation, good biodegradability and antioxidant [49,50] and antibacterial properties [44,51]. Thus, it is a good candidate for producing recyclable food packaging materials. However, compared with PVA, polyvinyl chloride (PVC) and other synthetic packaging materials, CS still has some deficiencies due to its poor mechanical properties [52]. Some studies have been conducted on the improvement of the properties of chitosan. On the one hand, increasing the amount of natural dye can improve the characteristics of the UV light barrier, heat stability, antioxidant ability and pH sensing capacity [53]. On the other hand, adding an appropriate amount of chitin nanofibers to chitosan film can significantly enhance some properties, such as tensile strength (TS), waterproof ability and roughness of the film. However, when the amount of CN was further increased, the mechanical properties of the films were not further improved [54].

#### 4.4.2. Starch

Starch has the characteristics of colorless, tasteless, non-toxic, biodegradable, low cost, rich raw materials and good thermoplastics. It is considered to be the most promising natural source for packaging production [55]. Nevertheless, due to the poor mechanical and thermal properties of starch, it usually lacks strength and processing ability. Thus, adding a certain proportion of PVA, tara gum, cellulose and so on to starch can improve the service performance [23,26,35,56]. Another study shows that the intelligent starch/PVA film has certain antibacterial activity so that it can improve the preservation of fish [57].

#### 4.4.3. Gum

Gum is generally used in the food industry and is mainly present in plants, microorganisms and so on. Previous studies have shown that gum could improve some properties of smart films, such as film formation properties, tensile strength and permeability, and reduce the solubility, slip and heat resistance of the film. However, in packaging applications, there are many limitations, with poor water resistance, unstable mechanical properties and low antioxidation and antibacterial capacity. Therefore, research has been conducted to combine these gums with cellulose, cellulose analogues, potato starch, purple sweet potato and so on to prepare pH-sensitive films based on different colorants [58,59]. In some instances, Liang et al. revealed that added red cabbage extract into Artemisia sphaerocephala Krasch. Gum could decrease the tensile strength, light transmittance and transparency of the film, while increasing the elongation at break and moisture permeability [60]. Similarly, another study showed that when red cabbage extract was added to ASKG, the tensile strength, light transmittance and transparency of the film decreased, while the elongation at break and moisture permeability increased [60]. Moreover, other studies have shown that tara gum, gellan gum, agar and so on are also used in fish freshness monitoring, and they are generally used as a part of dye carrier components, and the addition of gum can increase tense strength values [61].

## 5. Active Sensor

Indicators use visual color changes to assess fish freshness, and active sensors constitute a system used to measure physical values and convert them into a signal that can be read by an instrument or observer. The most important properties of these sensors are their selectivity, sensitivity and response time [62].

### 5.1. Gas Sensors

In most studies, the target gas group monitored by a gas sensor is mainly TVB-N, which is one of the most efficacious and monitors the freshness indicators of fish expediently. Gas sensors are valuable tools for research because they are non-destructive and can provide rapid detection. Among them, metal oxide sensors are the most-studied gas-sensitive material so far. The use of gas sensors based on metal oxide semiconductor materials has many advantages, such as excellent sensitivity, low cost, convenience and fast detection process, and they play an important role in the speedy and nondestructive detection of the freshness of fish [63,64].

For example, Liu et al. designed a method for detecting a TMD α-Fe_2_O_3_ nanosphere gas sensor, which can rapidly evaluate the freshness of fish [65]. The nanosphere sensor reacts with TMA gas in only one minute, and the recovery time is less than four minutes. In addition, there is a perfect linear relationship between the response value displayed by the sensor and the TMA concentration. Similarly, Chang et al. prepared an amine sensor that can rapidly detect ammonia, DMA, TMA and other gases [66]. Traditional TVB-N analysis takes approximately four hours for sample pretreatment, but this sensor can detect volatile amine gas in only one minute. In addition, TVB-N analysis is not extremely sensitive to tilapia with low TMAO during storage, and amine sensors can also aptly reflect deterioration. The sensor system is mainly composed of core sensor components and concise flow channels. It has the advantages of low manufacturing cost, simple operation and wide application. Consumers can directly use it at home to detect the real-time freshness of fish.

### 5.2. Biosensor

Biosensors are devices that measure chemical or biological reactions and convert them into electrical signals. They are made up of three main components [67]:Sensor substrates are composed of materials such as polymers, glass, metals and even paper combined with biological receptors.Biological receptors are bound to sensors by physical or chemical immobilization methods.Elements of conversion.

These devices are functionally integrated and provide sensors through biometric elements that can detect the presence of specific analytes in the reaction. The signals produced by the sensor cause changes in the electrical or electronic output, which are converted by the sensor into measurable signals. Finally, the detector amplifies and processes the processed signal [68].

Hypoxanthine (Hx) is a sensitive indicator of the fish products’ freshness. In general, some classical methods for measuring Hx such as HPLC [69], chromatographic measurements [70] and so on are time-consuming, costly and inconvenient to operate. In order to solve the problems of traditional methods, researchers have developed a series of biosensors to determine Hx value, such as electrochemical fluorescence and visual sensors. Chen et al. developed a fluorescent biosensor based on the peroxidase activity of platinum nanoparticles (Pt NPs) and had a good application prospect in rapid and accurate evaluation of fish freshness [63]. The fluorescence intensity of the sensing system has a certain linear relationship with the Hx value. This biosensor can cope with various interfering substances and has good selectivity, convenience, low cost and reusability. After three consecutive experiments, the enzyme activity efficiency of nanoparticles is only reduced to 91%. Besides, Rong et al. [70] developed a paper biosensor utilizing the enzymatic conversion of Hx by xanthine oxidase (XOD) and then measured the release of the fish degradation index Hx. Compared with commercial kits, these biosensors are more cost-effective, do not need to add foreign reagents and are easy to carry.

### 5.3. Electronic Sensory Techniques

#### 5.3.1. Electronic Nose and Tongue

An electronic nose is a gas sensing system constructed by imitating the function of a human nose. It is mainly composed of three components: A series of gas sensors with wide selectivity that can determine volatile compounds, a signal preparation system and a pattern recognition system [71]. The electronic nose has the characteristics of high sensitivity, reusability, high efficiency and low cost. It can be used to measure and monitor the smell anywhere and has a wide range of applications. For example, the electronic nose is usually used in environmental, medical and pharmaceutical fields [72]. The equipment has been proven to be very effective in food industry applications, such as quality control, process monitoring, freshness assessment and authenticity evaluation [73,74,75].

Various electronic noses, such as electrochemical gas, metal oxide and conductive polymer sensors, have been used to assess fish freshness because of their different sample extraction and data processing methods. Research showed that a simple and reusable electronic nose made of a metal oxide semiconductor could be used to determine the freshness of hairtail fish in supermarkets. Based on the volatile amine produced by the sample in head space, two characteristics were extracted from the instantaneous response of each sensor by the principal component analysis (PCA) method. The results revealed that the electronic nose and TVB-N had a good correlation coefficient [76].

The electronic nose is suitable for gaseous systems, and the electronic tongue can obtain the rapid evaluation of complex liquids by identifying tastes in soluble foods. The electronic tongue is a device made of various sensor units by signal conversion. Ruiz-Rico et al. learned from the study that the Voltaire electronic tongue can determine the quality of cod during frozen storage [77]. The electronic tongue can successfully distinguish the number of days experienced by the sample, and the effect of the electronic tongue in detecting solids such as fish is worse than that of liquid. Since the freshness of fish is determined by the biogenic amines produced during the monitoring of fish corruption, Rodriguez-Mendes et al. [78] prepared a traditional volt electronic tongue based on a carbon paste electrode and a series of screen-printed electrode systems. It has been proven that both systems can monitor the deterioration of fish and send signals related to the formation of biogenic amines during storage.

#### 5.3.2. Colorimetric Sensor Array and Colorimetric Systems

A color sensor array uses a variety of sensors to evaluate the freshness of fish. Its principle is to combine gas with metal collagen to change color. These adhesives are dyes, and the bonding methods are mainly pi-pi bonds, metal bonds and hydrogen bonds. The absorption peak changes after binding, and all changes occur in the visible range. The image acquisition device extracts the sensor array image, and then processes the image to obtain the quantization result [4]. Therefore, the design of the color sensor array needs to meet two requirements:The dye must contain a center when participating in the reaction in order to strongly interact with the analyte.This interaction center must be closely coupled to strong chromatography.

Sensor arrays consisting of sensitive chroma dyes that mimic human olfactory functions have been developed [79]. The dye can change color according to the concentration of TVB-N, so as to intuitively evaluate the freshness of fish. For example, sensor arrays have been successfully applied to monitor fish freshness. The self-made sensor is used to detect fish fillets in different storage periods. The results show that the detection efficiency of this sensor is good [80]. Morsy et al. designed a sensor array that contains 16 chemically sensitive units. These substances are typical corruption-inducing compounds, such as TMA, DMA, cadaveric alkali, putrescine, etc. [81]. Computer vision technology can predict the freshness of fish quickly and at a low cost. At room temperature, the shelf life of frozen cod fillets is monitored by a colorimetric sensor array composed of a porphyrin compound and an acid–base indicator [82].

### 5.4. Other Active Sensors

Capacitive sensors have the advantages of low temperature dependence, good thermal stability, low power requirements, simple construction, high sensitivity and low cost [83]. In addition, the circuit signal of the sensor is easy to process and can selectively detect specific gases. Among capacitive sensors, the sensor with a metal insulator semiconductor (MIS) structure has a better monitoring effect. However, in previous papers, MIS sensors are relatively few in the study of monitoring fish freshness [84]. Senapati et al. prepared an MIS sensor with Au, which can rapidly and accurately determine the freshness state of fish [85]. The manufacturing principle of the MIS structure sensor is based on a silicon substrate, with Ag-SnO_2_ as the sensing material of the SiO_2_ layer and Au as the metal electrode. This sensor technology can respond to raw fish in only four minutes. Compared with the traditional methods of monitoring TVB-N and TVC, this method has higher freshness status, requires less time and involves a simpler experimental process.

On the other hand, Nguyen et al. used the air pressure increment caused by the gas emissions of rotten fish to reflect the freshness of fish [6]. This air pressure sensor has high-resolution characteristics. It only needs micro watt power to accurately measure air pressure. Another interesting phenomenon in this experiment is that the sensor tag obtains energy and transmits data from a smartphone through NFC. The pressure value is used to categorize fish freshness into different levels on the phone screen, so as to ensure the freshness of the fish through the NFC smartphone.

## 6. Nuclear Magnetic Resonance Spectroscopy

Nuclear magnetic resonance (NMR) spectroscopy is one of the most common techniques used by researchers to identify molecular structures and explore chemical reaction processes. It is based on atomic spin and an odd number of neutrons or protons, such as hydrogen. The fish samples are placed in a magnetic field, and they are excited by radio waves, causing the nuclei to vibrate, which are then detected by sensitive radio receivers. The NMR instrument uses electromagnetic waves in the radio frequency range to interact with fish samples. This means many NMR techniques require smaller samples (milligrams, even micrograms), are noninvasive and nondestructive to samples, allow fast and safe operation and inflict no pollution on the environment. Nevertheless, the sample concentration can have an impact on the NMR results, and high concentrations can reduce the resolution due to saturation or increased viscosity. According to the adjusted parameters, this method can be used for the rapid analysis of fat, water and protein. Additionally, NMR may be a better selection for analyzing fish freshness. Therefore, the application of NMR in the fish industry and research can gradually replace traditional technology. However, NMR has the characteristics of expensive and complex equipment and high requirements for operators, which limit its wide application [6].

The applications of NMR techniques in fish research can be divided into three types: Magnetic resonance imaging (MRI), low-field NMR (LF-NMR), and high-resolution NMR (HR-NMR) [86]. Among them, HR-NMR is mainly used in the certification of fish oil and lipids and appears to be able to distinguish between different species, geographical origin, production and process histories. In HR-NMR, ^13^C-NMR is used to act on specific molecular sites to determine the isotopic content of fish, which has a significant contribution to the application of food certification [87]. At the same time, it also provides a new perspective for detecting fish freshness.

Heude et al. proposed a method based on ^1^H HR-NMR spectroscopy, which can rapidly determine the K-value and TVB-N content of fish (approximately 10–15 mg), and the results obtained are consistent with the cumbersome methods traditionally used to measure these two parameters [88]. The main advantage of the ^1^H HR-NMR method is that the K-value and TVB-N content can be determined directly on unprocessed fish samples without pre-extraction. Zhao et al. prepared the composite coating on tilapia fillets to study whether the coating can prolong the shelf life of tilapia fillets [89]. In this experiment, HR-NMR spectroscopy and PCA were used to take early metabolites as a biological index of fish freshness, and the effect of the coating solution on the changes of metabolites during storage was studied. The NMR analysis provides complete information on the changes of different metabolites during storage and the effects of metabolites on coatings. PCA was used to analyze major metabolites. Moreover, the inhibitory changes of pH, TVB-N and K-values further verified the NMR results. A summary of recent NMR techniques for fish freshness monitoring is offered in Table 2.

## 7. Optical Spectroscopic Techniques

With decreasing instrument prices and improvements in equipment and chemometric tools, spectroscopy has become a progressively attractive analytical technique for assessing fish freshness. The main advantages of using optical spectroscopic techniques are fast data acquisition, the simultaneous determination of multiple quality parameters and the ability to replace expensive and time-consuming reference techniques. In recent years, the qualitative and quantitative analysis of spectroscopic techniques, such as fluorescence spectroscopy, infrared spectroscopy and near-infrared spectroscopy in detecting fish freshness, has received extensive attention from researchers. As is shown in Figure 3, when the fish tends to spoil, the internal substances change, and these substances can be used to evaluate the freshness of the fish under the characterization of the spectrometer.

### 7.1. Fluorescence Spectroscopy

Fluorescence spectroscopy (FS) is a fast and non-destructive method that can screen a great number of various foods. Because of its high sensitivity and specificity, this technology has been widely used in food detection [96]. The principle of FS is based on the physical response of fluorescence and the interaction of substances called “fluorescein” in fish samples, including aromatic amino acids and nucleic acids, tryptophan, tyrosine and phenylalanine in proteins, as well as nicotinamide adenine dinucleotide (NADH) and many other low-concentration compounds. They receive energy in the form of electromagnetic waves (typically 180–800 nm) from the external environment and emit this energy in the form of higher frequency light. The energy is collected by optical probes or dedicated spectrofluorometers [68]. When excitation light illuminates a high-concentration sample, fluorescence is generated near the excitation light entrance, but this fluorescence does not enter the fluorescence detector. In addition, FS uses microwave digestion technology during pretreatment, which can reduce the time required for sample processing and ensure a complete reaction.

In recent years, synchronous fluorescence spectroscopy (SFS) has shown tremendous potential in various applications. It can obtain the entire fluorescence landscape and retain the data related to multiple fluorescent lamps, while the classical emission spectrum is mainly based on single fluorescence. The excitation emission matrix (EEM) is a set of fluorescence spectra obtained at continuous release wavelengths to create three-dimensional diagrams. This method has been widely used in the non-destructive measurement of the physical and chemical performance of food. In fish freshness evaluation, EEM is applied to estimate the freshness index and ATP content of frozen fish with high accuracy. Nevertheless, this method is a point measurement, which can only estimate the quality to a certain point, and the freshness of other parts of fish cannot be understood.

According to the partial least-squares regression (PLSR) regression model, ElMasry et al. explored EEM spectra in two different studies, both evaluating mackerel samples [96,97]. The freshness values were determined by the HPLC method and used to construct the regression model, with an R^2^ of 85% for the first task and 94% for the second task, using an optical probe to obtain sample excitation spectra to obtain R^2^ between 250 and 800 nm.

The traditional right-angle fluorescence spectrum can only measure the solution with absorbance less than 0.1. At higher absorptivity, the fluorescence intensity decreases due to the internal filter effect, and the emission spectrum will be wrong. In order to address this problem, front-face fluorescence spectroscopy (FFFS) was developed when only the material surface was examined. Lately, FFFS has been applied to monitor the freshness of the whiting fish fillets stored in vacuum packaging. The FFFS emission spectra of the same samples were scanned and used tryptophan residues (excitation: 305–450 nm, emission: 290 nm) and Schiff base (excitation: 400–600 nm, emission: 360 nm) as fingerprints. By concatenating spectral, physicochemical and instrumental datasets, 100% correct classification was obtained [98]. Likewise, Hassoun et al. [99] studied the potential of FFFS as a fast method that determines the freshness of whiting fillets stored at 4 °C for 12 days. Their study showed that the emission spectra of tryptophan (excitation: 290 nm, emission: 305–450 nm) and NADH (excitation: 340 nm, emission: 360–600 nm) can be regarded as fingerprints in order to assess the freshness of fish. The best results were obtained based on the fluorescence emission spectra of tryptophan and NADH linked with physicochemical parameters, with an excellent correct classification ratio of 90.48%.

### 7.2. Infrared Spectroscopy

The chemical bonds present in fish organic matrices vibrate at specific frequencies that depend on the mass of the constituent atoms, the shape of the molecules, the stiffness of the bonds and the period of the associated vibrational coupling. Based on the above characteristics, infrared (IR) spectroscopy has become a common analysis method for simultaneous, rapid, and nondestructive determination of major components in fish. The sample dosage of the IR spectrometer is generally between 1 and 2 mg, and the light transmittance is generally maintained in the range of 10% to 80% when taken by the tablet method. Since the humidity of the test sample will affect the transparency of the sheet, the test sample should be dried under infrared light first.

Recent advances have triggered the further extension of IR technology to Fourier transform infrared (FTIR) spectroscopy. Notably, FTIR spectroscopy serves as a reliable, accurate and rapid method to assess the freshness of salmon fillets in real-time at different packaging temperatures and atmospheres [100].

### 7.3. Near-Infrared Spectroscopy

Among the various methods for evaluating food freshness, near-infrared (NIR) spectroscopy has the characteristics of simplicity, rapidity and non-destructiveness, and has been widely accepted by researchers. Some human-perceived spoilage changes such as exterior color change, odor generation, mucus formation and detectable sensory spoilage are the result of decomposition and metabolite formation caused by microbial growth. Visible and NIR (Vis/NIR) spectroscopy is of great interest because spoiled fish contain functional groups such as C–H, N–H and O–H, which are closely related to overtones and combined vibrations in the Vis/NIR spectral region.

An expanding array of studies have indicated the application of NIR in fish products. For instance, researchers found that NIR spectroscopy uses biogenic amine content to quickly and accurately assess fish freshness (R^2^ = 0.98) [101]. In addition, researchers have used near-infrared spectroscopy in the 800–2500 nm region to predict the potential of bacterial populations in salmon stored at 4 °C. The PLSR model was built on 72 data points and predicted the total number of aerobic slabs for the calibration and validation datasets with R^2^ values of 0.95 and 0.64, respectively [102]. Similarly, Wu et al. [103] investigated the ability to use VIS/NIR to predict salmon cold storage time. At the same time, a double-layer stacking denoising self-encoding neural network (SDAE-NN) algorithm is introduced to establish a prediction model without spectral preprocessing. Compared with PLSR and back-propagation neural network (BPNN) algorithms, SDAE-NN can obtain a better coefficient of determination and smaller prediction error, R^2^ = 0.98, RMSEP = 0.93 days. Compared with traditional methods, this technique has the advantages of non-destructiveness, low cost, rapidity and no pretreatment, and is more effective in evaluating fish freshness.

In addition, according to the principle of Vis/NIR, researchers have developed a new technology called imaging spectroscopy, namely hyperspectral imaging (HSI). HIS technology has been used for rapid, non-invasive analysis of freshness and overall fish quality in three main application areas: Chemical and physical analysis (moisture content, protein and so on), monitoring of food processing (thermal processing, freezing and chilling, dehydration) and food safety assessment (freshness evaluation, defect detection) [104]. The advantage of the HSI system compared to other non-destructive techniques is that it can simultaneously detect fish surface damage and different quality attributes. However, this technique cannot be used to detect internal damage in fish because the depth of light penetrating the surface of the fish is limited to a few millimeters [105]. Shi et al. developed a reliable functional neural network and used the optimal wavelength of HSI to estimate the freshness of tilapia fillets [106]. The results showed that the TVB-N, TAC and K-value of tilapia fillets increased under various storage conditions, while the sensory score decreased with the increase in storage time. In order to simplify the model, the author also used the continuous projection algorithm (SPA) to select nine optimal wavelengths, and then constructed a mathematical model in the form of body temperature according to the selected wavelengths, TVB-N (R^2^ = 0.9973), TAC (R^2^ = 0.9982), K-value (R^2^ = 0.9898) and sensory evaluation (R^2^ = 0.9765) of tilapia fillets. The ability of functional models with HSI to predict fish freshness has been verified.

## 8. Challenges Posed by Emerging Approaches

Based on the above studies, it can be revealed that visual indicators derived from intelligent packaging, active sensors and spectroscopic techniques show great potential in fish freshness evaluation. These methods undoubtedly improve the level of monitoring and the scalability of observations. Compared to traditional assessment techniques, the innovative technique has been shown to be a non-destructive, cost-effective and rapid method for automated or online fish freshness monitoring. At the same time, there are some unavoidable problems that limit the application of these emerging technologies.

In general, some traditional methods are still very popular in the market industry as emerging methods are limited by selectivity, sensitivity, poor data analysis and complex calibration. The use of intelligent packaging based on chemically reactive natural colors is a promising strategy that can bring many benefits to the food industry by preventing fish waste and spoilage. However, there has been no perfect and stable indicator label in which the dye reacts to environmental factors over time. This instability is one of the issues that future research needs to address. In addition, biopolymers have certain disadvantages such as water sensitivity, swelling behavior and so on, which still remain to be solved. Most pH-sensitive labels developed are single-use materials, and commercialization costs are also a factor to consider.

Gas sensors, biosensors and electronic sensory technologies in active sensors are by far the most promising for entry into the market, especially the biosensors that are easy to carry. The biggest feature relative to visual indicators is their high sensitivity, but so far, they are not ideal in terms of stability. Since these sensors are easily affected by physical and chemical environmental factors, the detection results are biased. Real-time monitoring applications of biosensors in the fish supply chain are yet to be developed, although they can directly assess the chemical composition of fish during spoilage. Electronic nose and tongue technology has the advantage of fast detection (1–10 min); however, some of them require high operating temperatures of 250–500 °C and zero gas such as vacuum or N_2_, and response baseline values are also prone to skew after long runs. Furthermore, colorimetric sensor devices are not intelligent enough to be used on an industrial scale.

NMR may be a better choice for the analysis of heterogeneous samples such as fish. However, compared with other spectroscopic techniques, this technique has complex equipment and low sensitivity, especially the expensive basic equipment and high maintenance expertise, which limit its rapid development in the detection of fish freshness. None of these techniques have become commonplace in daily practice due to the relatively expensive and complex operation of spectroscopic instruments. In addition to limited applications, these instruments face several challenges. For example, fluorescence spectroscopy requires more research to identify the most efficient emission wavelengths, near-infrared spectroscopy always needs to be calibrated with reference methods based on fish under different storage conditions and hyperspectral imaging instruments collect too many data (captured data for the entire field of view and the entire space), thereby placing a heavy burden on the chip.

## 9. Conclusions and Future Trends

Nowadays, while sensory, physicochemical, microbiological and other traditional methods are time-consuming, laborious and expensive, they are used more as auxiliary means. It is worth noting that different types of fish have different microbial characteristics and produce various metamorphic products and by-products. Therefore, the characteristics of fish should be considered when choosing colorants in visual indicator applications. This characteristic makes polymeric materials more effective in sensing volatile components as the interaction is enhanced between the immobilized pigments and the chemical vapors, leading to better color resolution. Researchers have proposed unique methods and points of view to use NFC on mobile phones to make up for the inconspicuous color rendering characteristics of natural colorants. In addition, the sensitivity of the indicators depends on the category of the polymer in the matrix. For instance, biopolymers have a looser polymer chain structure than synthetic polymers, so they have a better capacity to transport gas and water vapor.

For active sensors, attention should be paid to improving their selectivity and sensitivity, simplifying the preprocessing process, improving their measurement accuracy and ensuring reproducibility, which can accelerate active sensors to replace traditional methods and be suitable for the market.

Spectroscopic devices can represent the best approach for industrial applications, are compact and do not require contact with fish. Due to the fast and less invasive acquisition process, these instruments are suitable for on-line industrial monitoring. With the continuous innovation of chemometrics software and informatics techniques, some limitations of spectroscopic techniques and instrumental sensors are being addressed. These emerging technologies overcome the shortcomings of traditional evaluation methods such as complex operation and strong subjectivity and increase the sensitivity and efficiency of detection. Among them, HSI technology combines the advantages of spectroscopy and imaging. Spectral data can illustrate the chemical composition and structure of the sample, while image information can reflect the spatial distribution, external properties and geometric structure of the sample.

At last, it is relatively one-sided to evaluate the edibility of fish by freshness. The fish is considered spoiled when there is a problem with the taste, even if the freshness is still within the standard range. At this time, it is necessary to use the quality to evaluate the edibility of the fish. In the future, a set of schemes suitable for all kinds of fish quality will be put forward, which requires further unremitting efforts of researchers.

## Figures and Tables

**Figure 1 foods-11-01897-f001:**
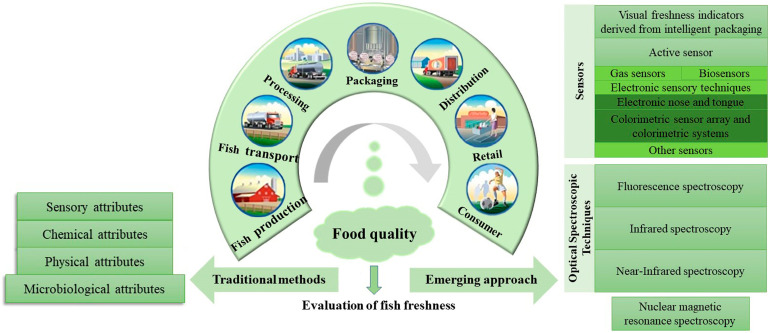
Graphical summary of the approach for fish freshness evaluation.

**Figure 2 foods-11-01897-f002:**
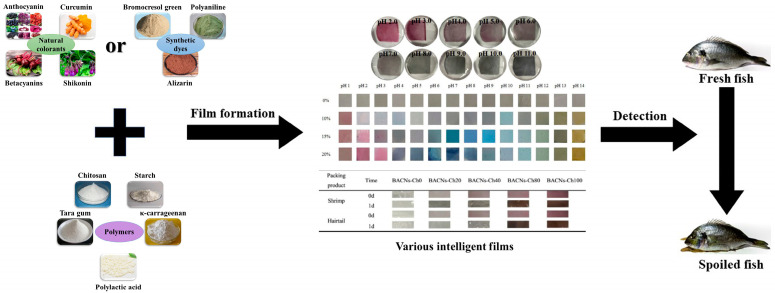
Manufacturing process and application of intelligent indicator.

**Figure 3 foods-11-01897-f003:**
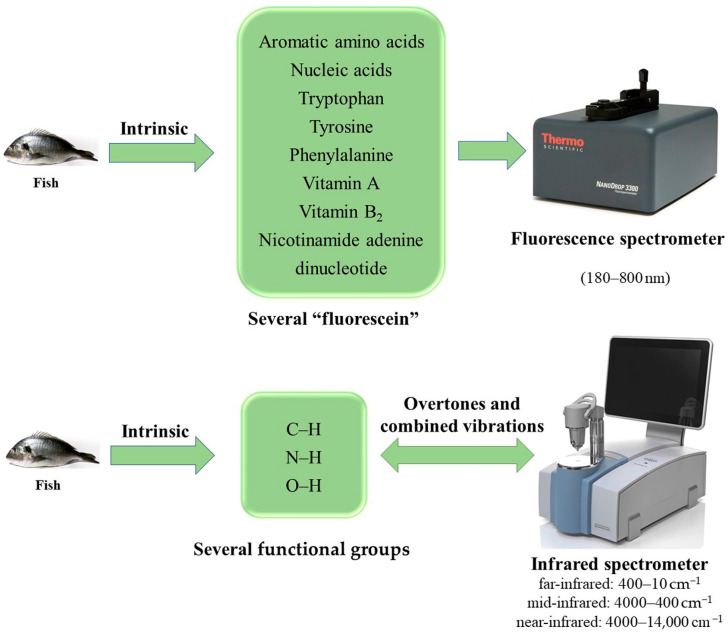
Characterization of fish intrinsic substances and spectroscopic instruments.

**Table 1 foods-11-01897-t001:** Natural colorant indicators and synthetic dye indicators for fish freshness monitoring.

Type of Pigment	pH-Sensitive Dye	Host Materials	Measurements	Color Change	Reference
Natural colorants	Anthocyanin	Oxidized chitin nanocrystals gelatin	TVB-N	Purple-gray blue or brown	[22]
	Anthocyanin	Starch polyvinyl alcohol and glycerol	TVB-N	Purple-green	[23]
	Anthocyanin	Carboxymethyl-cellulose and starch	TVB-N	Red-blue and green	[24]
	Anthocyanin	Sodium carboxymethyl starch and κ-carrageenan	TVB-N	Red-dark blue	[25]
	Anthocyanin	Bacterial nanocellulose	TVB-N	Red-gray	[26]
	Anthocyanin	Starch polyvinyl alcohol	TVB-N	Purple-gray	[27]
	Curcumin	Corn starch and polyvinyl alcohol	TVB-N	Yellow-red	[28]
	Curcumin	Gelatin and κ-carrageenan	TVB-N	Yellow-red	[29]
	Betacyanins	Glucomannan-polyvinyl alcohol	TVB-N	Purple-yellow	[30]
	Shikonin	Carboxymethyl cellulose and cellulose nanofibers	TMA	Pink-blue	[31]
Synthetic dyes	BCG	Sol-gel	TVB-N	Yellow-blue	[32]
	BCG	Porous anodic aluminum	TVB-N	Yellow-green	[33]
	BCG	Cellulose acetate	TMA	Yellow-blue	[34]
	polyaniline (PANI)	PANIi film tetraphenylethylene	TVB-N	Green-blue	[35]
	Alizarin	Starch-cellulose	TVB-N	Orange-reddish brown	[36]
	Alizarin	Zein nanofibers	TVB-N	Purple-magenta	[37]

**Table 2 foods-11-01897-t002:** Recent NMR techniques for fish freshness monitoring.

Methods	Chemometric Tools	Research Object	Main Results	Reference
2D ^1^H J-resolved NMR	Partial least squares discriminant analysis (PLS-DA) and orthogonal partial least squares discriminant analysis (OPLS-DA)	21 metabolites of intact zebrafish	Provide an efficient way for quality evaluation of semisolid and viscous foods	[90]
^1^H-NMR	Analysis of variance (ANOVA)	Amino acids, organic acids and alcohols	An alternative to the K-index or analogue indices	[91]
^1^H-NMR	ANOVA and PCA	Acyl groups, phospholipids and cholesterol	Evaluate differences in lipids composition	[92]
NMR	ANOVA, PCA and principal response curves (PRC)	Inosine, hypoxanthine, lactate, taurine, creatine, and TMA	The freezing-thawing cycles favored the increase of endogenous and exogenous enzymatic activities	[93]
NMR	PCA	25 metabolites of salmon	Detect high-added value compounds in salmon	[94]
^1^H HR-MAS NMR	ANOVA and PCA	The K-value and the TVB-N concentration	Allow a direct measurement of these two parameters directly on unprocessed fish	[88]
NMR	OPLS-DA and PCA	42 metabolites of tilapia fillets	Extend the applicability for fish by-products analysis	[89]
^13^C NMR	PCA and Bayesian belief networks (BBN)	Muscle lipids of various cod	Correct classification of 78% of samples belonging to the different species	[95]

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
