# Peer review of "Emerging Approach for Fish Freshness Evaluation: Principle, Application and Challenges"

_foods, 2022, doi:10.3390/foods11131897_

Round 1

Reviewer 1 Report

The manuscript "Emerging approach for fish freshness evaluating: principle, application and challenges" is within the scope of the Foods MDPI journal and should be of interest to its readership. Overall, the manuscript presents reviwed results on methods of fish freshness evaluation, and they describe several practical challenges.

However, authors need to highlight what the paper brings as a novelty in fish science and not only reviewed the literature, especially when describe food science techniques on fish quality.

Authors should compared traditional methods (e.g. appearance, color, meat elasticity or texture, odor, and taste of the fish) versus emerging approaches (e.g. NMR spectroscopy and sensors).

Authors should describe the limitations of current technologies. Please, add a separate subtopic including the limitations of reviewed studies.

The NMR spectroscopy should be better described, especially those presented in the Tables. Please, add more details regarding the reviewed studies. Tables need to be improved, as they are confusing (please, define all acronyms and technical terms in the Tables 1 and 2).

Considering the volume of data the authors reviwed, I believe the discussion could be significantly improved. Many results could be better explored as Tables or Figures. Moreover, considering that it is a literature review, the Figures may be more useful. Figure 1 and Figure 2 are suitable. Authors may include one or more additional figures. 

Reviewer 2 Report

 Emerging approach for fish freshness evaluating: principle, application and challenges

General comments

This topic is interesting and fit in the scope of the journal. In the current scenario, the update on the recent technologies on fish freshness estimation is very important. Authors have clearly highlighted the application of different techniques on the fish freshness measurement. However, the following specific points should be considered.

Specific comments

Abstract

Abstract looks very general. Add a specific points such as mechanism of non-destructive techniques and how it will interact/useful for fish freshness evaluation

Keywords: Avoid the words used in the title

Introduction

L32: What are all the traditional techniques used for fish freshness evaluation? Explain

L59: Write the novelty of this review

Write the application of image processing/mobile based methods for fish freshness detection

Write the methodology section including search engine used for collection of articles, keywords used, parameters considered to shortlist the article, total number of articles collected, etc.,

Add the images of the fish figures (fresh and spoiled fish) for better understanding to the readers

L166: Explain the reason behind the color change of spoiled fish as compared to fresh fish?

Explain the challenging aspects associated with active sensor

Write the challenges associated with Electronic Nose and Tongue during fish freshness evaluation

Write the molecular interaction of spectroscopic techniques with fish samples and correlate how it was used for fish freshness detection

Write the effect of sample size (calibration) in the accuracy of the spectroscopic techniques

The recent applications of spectroscopic techniques (FT-IR, FT-NIR, etc.,) are highlighted in the following papers. Refer these papers.

https://doi.org/10.1016/j.tifs.2021.12.021​

Write the effect of different pre-processing techniques on the accuracy of spectroscopic method during fish freshness measurement

Please remove the old references (Published before 2010)

Reviewer 3 Report

In the Abstract the conclusion drawn by the authors should be presented.

“As presented in Figure 1.” Is not a full sentence

Fluorescence spectroscopy should also show signals from protein (die to tryptophan, phenylalanine and tyrosine) and most probably also from NADH.

Round 2

Reviewer 2 Report

The manuscript as an interesting topic that has practical relevance as well. Authors have revised the manuscript thoroughly according to reviewers' comments and suggestions. The revised manuscript has a logical structure. The overall scientific quality of the manuscript has improved significantly due to the revision. Authors gave detailed answers for reviewers' questions. I accept all answers and modification made by the authors.